

# Density of cannabis outlets *vs.* cannabis use behaviors and prevalent cannabis use disorder: findings from a nationally-representative survey

Wit Wichaidit[1,2], Ilham Chapakiya[3], Aneesah Waeuseng[3], Kemmapon Chumchuen[1] and Sawitri Assanangkornchai[1,2]

[1] Department of Epidemiology, Faculty of Medicine, Prince of Songkla University, Hat Yai, Songkhla Province, Thailand
[2] Centre for Alcohol Studies, Hat Yai, Songkhla Province, Thailand
[3] Division of Computational Science (Statistics), Faculty of Science, Prince of Songkla University, Hat Yai, Songkhla Province, Thailand

## ABSTRACT

**Background:** Thailand recently decriminalized (*de facto* legalized) cannabis use and sales. However, nationally representative data are scarce with regard to cannabis use behaviors and its association with cannabis outlet density. The objectives of this study are: (1) to describe the prevalence of cannabis use behaviors and cannabis use disorder among the general adult population of Thailand; (2) to describe the extent that the density of cannabis outlets is associated with cannabis use behaviors, cannabis use disorder, and the amount of cannabis smoked per day.

**Methods:** We conducted a community-based cross-sectional study in 11 provinces and the Bangkok Metropolitan Area. Participants were residents of sampled communities aged 20 years or older. We requested literate participants to self-administer the questionnaire and interviewed participants who could not read. We analyzed data using descriptive statistics with sampling weight adjustments and multivariate logistic regression analyses.

**Results:** The prevalence of current cannabis use was 15 percent. At a 400-m radius, participants who reported three cannabis outlets had 4.2 times higher odds of being current users than participants who reported no outlet (Adjusted OR = 4.82; 95% CI [3.04–7.63]). We found no association between outlet density and hazardous cannabis use or cannabis use disorder, nor association with the amount of cannabis use among cannabis smokers.

**Discussion and Conclusion:** The patterns of association between outlet density and cannabis use behaviors were inconsistent. Furthermore, limitations regarding outlet density measurement and lack of temporality should be considered as caveats in the interpretation of the study findings.

Corresponding authors
Wit Wichaidit, wit.w@psu.ac.th
Sawitri Assanangkornchai, savitree.a@psu.ac.th

# INTRODUCTION

Approximately 219 million people around the world consume cannabis (*Elflein, 2023*), 10% of whom are affected by cannabis use disorder which impairs normal social functions (*Connor et al., 2021*). Previous studies have found positive associations between cannabis outlet density and cannabis use behaviors (*Lankenau et al., 2019*; *O'Hara, Paschall & Grube, 2023*; *Pedersen et al., 2021*; *Rhew et al., 2022*) and cannabis use disorder (*Mair et al., 2021*). However, these studies were conducted in locations where a centralized database of licensed and unlicensed cannabis dispensers was available. We hypothesize that such associations are also found in low and middle-income countries. In that regard, low and middle-income countries that have recently decriminalized or legalized cannabis use may not have such data infrastructure, requiring an alternative measurement of cannabis outlet density. Furthermore, previous studies have not assessed the extent that outlet density was associated with the dosage of cannabis used (in grams) per day of use. As the number of outlets implies availability, which is associated with a higher prevalence of heavy cannabis use (*Pedersen et al., 2021*), we hypothesize that there is a positive association between cannabis outlet density and cannabis dosage. As frequency and dosage of use are associated with cannabis use disorder (*Callaghan, Sanches & Kish, 2020*), we hypothesize that there is a positive association between cannabis outlet density and cannabis use disorder.

Thailand is a middle-income country in Southeast Asia. After several years of gradual introduction of medical cannabis use as part of Thai Traditional Medicine, in February 2022, the Royal Thai Government issued the Notification of Ministry of Public Health, RE: Narcotics under Category 5 of the Narcotics Act B.E. 2565 (2022). The Notification removed extracts of Cannabis family plants with THC content of no more than 0.2 percent by weight, as well as extracts from cannabis or hemp seeds that were domestically grown, from the list of illegal narcotics (*Royal Thai Government, 2022*). In effect, the Notification allowed for legalized and relatively unrestricted sales of cannabis to any person aged 20 years or older (*CBS News, 2023*; *Tang, 2023*). This has led to concerns regarding a potential increase in cannabis use disorder and other drug abuse in the general population of Thailand.

Although previous assessments on cannabis use have been made in Thailand (*Assanangkornchai et al., 2022*; *Kalayasiri & Boonthae, 2023*), these assessments were made prior to the legalization. Timely and nationally-representative data with adequate detail to contextualize cannabis use may still be needed to support the decision of policy-makers. Data on the association between cannabis outlet density and cannabis-related behaviors and health outcomes can enable more evidence-based decisions regarding cannabis access and availability restrictions among relevant stakeholders. Thus, the objectives of this study are: (1) to describe the prevalence of cannabis use behaviors and cannabis use disorder among the general adult population of Thailand; (2) to describe the extent that the density of cannabis outlets is associated with cannabis use behaviors, cannabis use disorder, and the amount of cannabis smoked per day.
## MATERIALS AND METHODS

### Study design and setting

Community-based cross-sectional study in randomly selected provinces and sub-districts in Thailand.

### Study participants

The target population in this study included general population who resided in Thailand aged 20 years or older and had lived in the sampled study areas for at least 3 months. Inclusion criteria were: (1) being a member of the sampled households (having resided in the sampled household for at least 90 nights per year; (2) aged 20 years or older; (3) able to communicate in Thai language. Exclusion criteria were: (1) employees of the household; (2) tenants of the household; (3) institutionalized individuals (*i.e.*, temples, prisons, juvenile detention, hospitals, boarding schools, military bases, police bases).

### Sample size calculation

We calculated the sample size based on the assumption that the prevalence of cannabis use in Thailand was similar to the global prevalence of 2.5 percent (WHO Cannabis 2022), with 1 percent margin of error, at a 95% level of confidence, and assuming design effect of two and that 85% of sampled residents agreed to participate in the study. We subsequently obtained a sample size of 2,158 individuals. We then randomly sampled 11 provinces plus Bangkok (the capital) and assigned the number of samples in each region according to probability proportional to size.

### Study instrument

Our study instrument was a questionnaire that included four sections, namely: (1) Characteristics of the study participants; (2) Patterns of cannabis use (cannabis use, impacts of cannabis use, cannabis use disorder screening questionnaire/CUDIT-R, kratom use, alcohol use, tobacco use excluding electronic cigarette); (3) Access to cannabis and exposure to cannabis, exposure to secondhand cannabis smoke, opinions regarding cannabis control policy, and attitude and norms regarding cannabis use; and (4) Other health behaviors.

### Study variables

*Exposure (Density of cannabis outlets within given distances)*: The term "outlet" in this study refers to the point at which goods are sold or distributed, thus places that either cultivated or sold cannabis had the potential to distribute cannabis. We included four measurement questions, each with regard to the number of outlets at varying proximity. The first question was "C1_6. As you are aware, within 400 m from your home (approximately 5 min' walk), how many cannabis growing and selling points are there?". Similar questions were asked for the 800 m (10 min' walk), 1,200 m (15 min' walk), and 1,600 m (20 min' walk). One of the investigators (WW) developed the questions based on his own understanding regarding travel time on foot by walking at a moderate pace.

*Outcome 1 (Cannabis use behaviors)*: We measured cannabis use by adapting a questionnaire previously used in a study in Thailand (*Assanangkornchai et al., 2022*) in combination with the Daily Sessions, Frequency, Age of Onset, and Quantity of Cannabis Use Inventory (DFAQ-CU) (*Cuttler & Spradlin, 2017*). One investigator (WW) translated the English-language questionnaire to Thai, then used machine translation to back-translate into English. Points of discrepancies were identified and the Thai translation was further revised and finalized. We assessed cannabis use behaviors with two questions: (1) "B1_1. In this lifetime, have you ever used cannabis?", and; (2) "B1_5. When was the most recent occasion that you used cannabis?". We considered participants who answered "No" to the lifetime use question to be never users. We considered those who answered "Yes" to the lifetime use question but answered "More than 1 year ago" to be former users, and the remaining to be current users. We treated those who refused to answer either of the two questions, and those who said "Don't know" to the question on the most recent occasion as having missing data.

*Outcome 2 (Cannabis use disorder)*: We used the Cannabis Use Disorder Identification Test-Revised (CUDIT-R), a screening instrument, to measure potential cannabis use disorder in this study. The 8-items screening test had a range of 0–24 points, with the cut-off at 8–11 points for hazardous use, and 12–24 points for cannabis use disorder (*Adamson et al., 2010*).

*Outcome 3 (Amount of cannabis used (grams) per day of use)*: Amount of cannabis used per day of use was calculated based on the response to the questions: (1) "B1_12a. In each cannabis use session, in general, how many grams of cannabis do you use yourself?" and; (2) "B1_12b. On the day that you use cannabis, in general, how many times do you use cannabis per day?" After we treated the "don't know" and "refuse to answer" of each question as NA, we multiplied the two numeric responses in order to obtain the approximate amount of cannabis used by the participant (in grams per day of use).

*Covariables (General characteristics of the study participants)*: We used the same set of questions as those used in previous rapid community-based surveys on alcohol consumption during the COVID-19 pandemic (*Vichitkunakorn et al., 2020*; *Wichaidit et al., 2022*). However, we modified the questions regarding the number of household members, gender identity, and personal/household income to suit the contexts of this study.

## Data collection

We contracted a survey research firm (SAB Co., Ltd., Delhi, India) to collect the study data based on the protocols, supporting documents, and the study instrument that have received ethical approval. We asked the survey research firm to use a stratified multi-stage cluster sampling technique to select our study participants. In Stage 1, we performed stratified random sampling by selecting two provinces in each of the five regions of Thailand (except for Bangkok Metropolitan Area, in which we selected Bangkok and one neighboring province). In Stage 2, the survey research firm purposively selected the main municipality of the Mueang (Capital) District of each selected province to represent urban area population, and randomly sampled a sub-district in another district to represent the

rural population (except for Bangkok, in which two urban districts were randomly sampled). In Stage 3, the survey research firm performed clustered sampling by contacting the health promotion hospital (primary care facility) responsible for the selected municipality area or sub-district, then randomly sampled village health volunteers who worked under each health promotion hospital using simple random sampling. The survey research firm then asked the village health volunteers to escort the data collectors to all households under the volunteer's care.

Upon arriving at each household, data collectors introduced themselves and informed all adults in the household who met the study criteria about the study and asked for verbal informed consent. Data collectors also inquired each participant about their ability to read. Among participants who were able to read, data collectors would give the tablet or smartphone containing the questionnaire to the participant to self-administer. Among participants who were not able to read, data collectors would collect data using face-to-face interviews. Before data collection began, data collectors would ensure that the health volunteer was not in the vicinity. If a participant was found to experience cannabis use disorder according to the CUDIT-R screening questionnaire, the data collectors would inform the participant accordingly and distribute an information leaflet titled "What You Need to Know about Cannabis Use Disorder" with an approval stamp from the Human Research Ethics Committee of the study university of the primary author (WW). The information leaflet contained basic facts about cannabis use disorder as well as contact information of substance abuse treatment centers in various regions of Thailand.

## Data management

The survey research firm's data management team oversaw the data collection process by collaborating with the team's supervisors to check the quality of the collected data each working day. The data management team also used a software package to prevent errors, *e.g.*, incorrect entry of numbers or data inconsistency. The data management team then performed additional data processing before sending the final data to the investigators for further analyses.

## Data analyses

We described prevalence of the three outcomes (cannabis use behavior, hazardous cannabis use, and cannabis use disorder) using descriptive statistics. We used weighted mean and standard errors to describe continuous variables and we used weighted percentage and standard errors to describe categorical variables. We described the association between cannabis outlet density and cannabis use behavior or hazardous cannabis use using bivariate descriptive statistics. We also used multivariate logistic regression to calculate adjusted odds ratio with 95% confidence intervals. We described the association between cannabis outlet density and the amount of cannabis used by cannabis smokers using bivariate descriptive statistics. We also used multivariate linear regression to calculate adjusted correlation coefficient (Betas) with 95% confidence intervals. We cleaned and analyzed data using the *epicalc* package in R (*Chongsuvivatwong, 2015*), and

used the *Survey* package (*Lumley, 2010*) to adjust for sampling weight and complex survey design.

For multivariate analyses, previous studies have found that cannabis use is associated with sex (*Fond et al., 2021*), tobacco smoking (*Fond et al., 2021*), marital status (*El Marroun et al., 2008*), religion (*El Marroun et al., 2008*), educational level (*El Marroun et al., 2008*). Meanwhile, cannabis use disorder is associated with sex (*Fond et al., 2021*), tobacco smoking (*Fond et al., 2021*), income (*Mair et al., 2021*), unemployment (*Mair et al., 2021*), and age of onset of cannabis use (*Schlossarek et al., 2016*). Thus, in multivariate analyses for our study outcomes, we adjusted for the participant's sex, age, tobacco smoking status, marital status, income, religion, occupation, educational level, and age of onset of cannabis use as confounders based on *a priori* identification according to the literature.

With regard to outlet density in particular, on all distances from 400 to 1,600 m, half of the participants answered "Don't know" to the number of outlets. We decided to recode "don't know" as 0 in order to maintain statistical power, but still treated the refusal to answer as missing values. For participants who did not indicate their age of cannabis initiation, we arbitrarily imputed the median age of initiation for those older than the median age. We imputed 20 years as the proxy age of initiation for participants who were younger than the median age. We also decided to collapse the "possible cannabis user disorder" group with the "hazardous use" group for the bivariate analyses and logistic regression analyses. We otherwise analyzed all data as they appeared.

In order to assess the extent that the re-coding could have introduced bias to the study findings, we also performed sensitivity analyses on the association between outlet density and the outcomes without the imputations, *i.e.*, we treated all "Don't know" responses regarding outlet density as missing values. For the association between outlet density and amount of cannabis used, our sensitivity analyses also included age of initiation without imputation.

### Ethical considerations

This study received ethical approval from the Human Research Ethics Unit, Faculty of Medicine, Prince of Songkla University (REC. 65-434-19-2). Investigators received waiver of written informed consent, which included waiver of the requirement to record the participant's name, based on the sensitive and politicized nature of cannabis use and trade in Thailand.

## RESULTS

The majority of 2,191 participants in our study, were female, married, with high school education or less, worked in manual labor and earned 15,000 THB per month or less (Table 1). Approximately 15 percent of our participants were former cannabis users, and another 15 percent were current users (used cannabis within the past 12 months prior to the survey). The mean age at initiation of cannabis use was 29 years. The main purpose of cannabis use was recreation followed by other non-medical purposes, and the most common method of cannabis use was ingestion (eating or drinking). Among current

cannabis users, 9 percent had a hazardous level of use, and 3% had possible cannabis use disorder. The majority of participants did not have a cannabis outlet within a 1,600-m radius (*i.e.*, one mile).

Cannabis outlet density was significantly associated with former and current uses (*vs.* never use), although the strength of the association varied by the radius (Table 2). Within the 400-m radius, participants who reported one cannabis outlet had 2.8 times higher odds of being current users than participants who had no outlet (Adjusted OR = 2.86; 95% CI [1.87–4.37]), whereas participants who reported three cannabis outlets had 4.2 times higher odds of being current users than participants who reported no outlet (Adjusted OR = 4.82; 95% CI [3.04–7.63]). However, the strength of the association appeared to decrease with the increase in the number of outlets. For example, participants who reported 1 outlet within a 1,600-m radius were 5 times more likely to be current cannabis users than those who reported no outlet (Adjusted OR = 5.39; 95% CI [3.56–8.19]). However, the odds was only 2.5 times higher among those who reported three outlets within a 1,600-m radius (Adjusted OR = 2.49; 95% CI [1.65–3.76]). We found no association between outlet density and hazardous or possible cannabis use disorder among current cannabis users (Table 3), nor did we find any association between outlet density and amount of cannabis used among current users (Table 4). Sensitivity analyses showed that the participants who answered "Don't know" regarding outlet density were less likely to be former and current cannabis users (Table S1). Sensitivity analyses also suggested that with regard to the association between cannabis outlet density and cannabis use status, re-categorizing those who answered "Don't know" regarding outlet density as people who reported no cannabis outlets in the proximity to their household biased most associations away from the null value (Table S2) but biased the association between cannabis outlet density and cannabis use amount towards the null value, with or without imputation of age at initiation as a confounder (Table S3).

## DISCUSSION

Using data from a nationally-representative survey, we described the extent that the density of cannabis outlets was associated with cannabis use behaviors, prevalent cannabis use disorder, and amount of cannabis consumed on each day of use within the past year. We found positive associations between cannabis outlet density and former and current use of cannabis, but no association between outlet density and hazardous or harmful use of cannabis or cannabis use amounts. The findings of this study have implications for stakeholders in behavioral health and public health. However, a number of considerations should be made in the interpretation of our study findings.

Up to 30 percent of our study participants indicated lifetime use of cannabis, although only half of the lifetime users were current users at the time of the survey. Our survey was conducted in May 2023, approximately one year after the decriminalization Notification took effect (*Royal Thai Government, 2022*). The prevalence in our survey was noticeably higher than the findings of national surveys during the early 2000s, when the War on Drugs was on-going (*Angkurawaranon et al., 2018*). The 15 percent of our participants who were former users at the time of study likely stopped using cannabis prior to the *de*

**Table 1 General characteristics of the study participants (*n* = 2,191).**

| Characteristic | Weighted percent ± Standard errors (unless otherwise indicated) |
| --- | --- |
| **Gender** | |
| Male | 47.4% ± 1.1% |
| Female | 52.6% ± 1.1% |
| Age in years (mean ± SE) | 45.8 ± 0.3 |
| **Marital Status** | (*n* = 2,187) |
| Single | 20.9% ± 0.9% |
| Married | 56.1% ± 1.1% |
| Co-habitation | 11.6% ± 0.7% |
| Widowed/divorced/separated | 11.3% ± 0.7% |
| It's complicated | 0.1% ± 0.1% |
| **Highest Level of Education Completed** | (*n* = 2,188) |
| Junior High School or Lower | 32.4% ± 1.0% |
| High School or Equivalent | 38.8% ± 1.0% |
| Some college (did not graduate) or currently studying | 2.8% ± 0.4% |
| Associate's degree | 11.8% ± 0.7% |
| Bachelor's Degree | 13.6% ± 0.7% |
| Graduate Degree | 0.6% ± 0.2% |
| **Occupation** | |
| Currently not working (Unemployed, Student, or Retired) | 5.8% ± 0.5% |
| Non-contracted occupation (Vendor, manual labor, farmer or fisher, or work or oneself such as food delivery person or selling things online) | 62.5% ± 1.1% |
| Public/private sector (civil servant or state enterprise, corporate employee, business owner, independent professional such as lawyer or architect, or others) | 31.7% ± 1.0% |
| **Personal Monthly Income** | (*n* = 2,183) |
| No more than 5,000 THB | 12.1% ± 0.7% |
| 5,001 to 9,999 THB | 19.4% ± 0.8% |
| 10,000 to 14,999 THB | 30.0% ± 1.0% |
| 15,000 to 19,999 THB | 17.0% ± 0.8% |
| 20,000 to 24,999 THB | 11.5% ± 0.7% |
| 25,000 to 29,999 THB | 5.4% ± 0.5% |
| 30,000 to 34,999 THB | 2.1% ± 0.3% |
| 35,000 to 39,999 THB | 1.3% ± 0.2% |
| 40,000 to 44,999 THB | 0.5% ± 0.1% |
| 45,000 to 49,999 THB | 0.0% ± 0.0% |
| 50,000 THB or more | 0.7% ± 0.2% |
| **Religion** | |
| Islam | 1.3% ± 0.2% |
| Buddhism | 97.8% ± 0.3% |
| Christianity | 0.9% ± 0.2% |
| **Smoking status (not including electronic cigarettes)** | |
| Never smokers | 70.7% ± 1.0% |
| Former smokers | 6.4% ± 0.5% |
| Current smokers | 22.9% ± 0.9% |

| Table 1 (continued) | |
|---|---|
| **Characteristic** | **Weighted percent ± Standard errors (unless otherwise indicated)** |
| **Cannabis use status** | ($n$ = 2,185) |
| Never users | 70.3% ± 0.9% |
| Former users (used but not in past 12 months) | 14.7% ± 0.7% |
| Current users (used within past 12 months) | 15.0% ± 0.8% |
| Age of initiation (years) for former and current users (mean ± SE) ($n$ = 651) | 29.1 ± 0.5 |
| **Initiated within the past year (only among former and current users) (% yes)** | 14.6% ± 1.4% |
| **Main purpose of cannabis use (among former and current users only)** | ($n$ = 651) |
| Recreation (for enjoyment/relaxation/socialization) | 59.5% ± 1.9% |
| Other non-medical purposes (appetite, sleep, others such as cooking or experimentation) | 34.2% ± 1.9% |
| Medical purpose (pain relief, cancer, seizures) | 6.3% ± 1.0% |
| **Method of cannabis use** | ($n$ = 651) |
| Ingestion (eaten or drank) | 60.1% ± 1.9% |
| Mixed with cigarette | 16.8% ± 1.4% |
| Rolled in a joint | 9.2% ± 1.1% |
| Smoked with water pipes | 5.2% ± 0.9% |
| Smoked in pipes or dried bong | 2.5% ± 0.6% |
| Others (infusion drop, capsules, spray, inhalants) | 6.2% ± 0.9% |
| **Cannabis use disorder status (among current users only)** | **($n$ = 273)** |
| Non-hazardous use (0–7 points) | 97.1% ± 1.0% |
| Hazardous use (8–11 points) | 1.1% ± 0.6% |
| Possible cannabis use disorder (12 points or higher) | 1.8% ± 0.8% |
| Cannabis use amount within past year (grams per person per day of use, current users who reported smoking cannabis only) ($n$ = 84) | 1.8 ± 0.2 |
| **Density of cannabis outlets within a given distance from place of residence**[*] | |
| **400 m (5-min commute)** | **($n$ = 1,750)** |
| None | 76.0% ± 1.0% |
| 1 outlet | 10.8% ± 0.7% |
| 2 outlets | 6.3% ± 0.6% |
| 3 or more outlets | 7.0% ± 0.6% |
| **800 m (10-min commute)** | |
| None | 72.0% ± 1.0% |
| 1 outlet | 11.3% ± 0.8% |
| 2 outlets | 6.3% ± 0.6% |
| 3 or more outlets | 10.4% ± 0.7% |
| **1,200 m (15-min commute)** | |
| None | 68.7% ± 1.1% |
| 1 outlet | 12.9% ± 0.8% |
| 2 outlets | 6.6% ± 0.6% |
| 3 or more outlets | 11.8% ± 0.7% |
| **1,600 m** | |
| None | 66.9% ± 1.1% |

(Continued)

| Table 1 (continued) | |
| --- | --- |
| **Characteristic** | **Weighted percent ± Standard errors (unless otherwise indicated)** |
| 1 outlet | 12.3% ± 0.8% |
| 2 outlets | 8.4% ± 0.7% |
| 3 or more outlets | 12.4% ± 0.8% |

**Note:**
* "Don't know" recoded as "None"; "Refused to answer" excluded from analyses.

**Table 2 Density of cannabis outlets within distances from participant's residence and cannabis use behaviors.**

| Distance and density | Never users | Former users | Current users | Crude OR (95% CI) for Former vs. Never users | Crude OR (95% CI) for Current vs. Never users | Adj. OR (95% CI) for Former vs. Never users* | Adj. OR (95% CI) for Current vs. Never users* |
| --- | --- | --- | --- | --- | --- | --- | --- |
| **400 m** | | | | | | | |
| None (n = 1,328) | 74.8% ± 1.1% | 14.0% ± 0.9% | 11.2% ± 0.8% | 1 (Ref.) | 1 (Ref.) | 1 (Ref.) | 1 (Ref.) |
| 1 outlet (n = 188) | 54.3% ± 3.6% | 21.7% ± 3.0% | 23.9% ± 3.1% | **2.14 [1.44–3.17]** | **2.93 [1.98–4.32]** | **2.68 [1.74–4.13]** | **2.86 [1.87–4.37]** |
| 2 outlets (n = 110) | 41.1% ± 4.7% | 16.3% ± 3.5% | 42.6% ± 4.7% | **2.11 [1.20–3.73]** | **6.91 [4.44–10.75]** | 1.66 [0.89–3.10] | **5.00 [3.13–7.98]** |
| 3 or more outlets (n = 120) | 47.5% ± 4.6% | 13.3% ± 3.1% | 39.2% ± 4.5% | 1.50 [0.84–2.67] | **5.50 [3.61–8.37]** | 1.59 [0.83–3.05] | **4.82 [3.04–7.63]** |
| **800 m** | | | | | | | |
| None (n = 1,247) | 75.2% ± 1.2% | 13.8% ± 1.0% | 11.0% ± 0.9% | 1 (Ref.) | 1 (Ref.) | 1 (Ref.) | 1 (Ref.) |
| 1 outlet (n = 196) | 52.2% ± 3.5% | 20.4% ± 2.9% | 27.5% ± 3.2% | **2.13 [1.43–3.18]** | **3.61 [2.48–5.24]** | **2.82 [1.82–4.35]** | **3.79 [2.52–5.71]** |
| 2 outlets (n = 110) | 44.7% ± 4.7% | 19.9% ± 3.8% | 35.3% ± 4.6% | **2.43 [1.43–4.12]** | **5.40 [3.42–8.52]** | **2.40 [1.31–4.39]** | **4.48 [2.72–7.39]** |
| 3 or more outlets (n = 179) | 49.2% ± 3.7% | 16.1% ± 2.7% | 34.7% ± 3.6% | **1.79 [1.14–2.81]** | **4.82 [3.33–6.97]** | 1.56 [0.94–2.59] | **3.91 [2.61–5.85]** |
| **1,200 m** | | | | | | | |
| None (n = 1,197) | 76.6% ± 1.2% | 11.9% ± 0.9% | 11.5% ± 0.9% | 1 (Ref.) | 1 (Ref.) | 1 (Ref.) | 1 (Ref.) |
| 1 outlet (n = 225) | 47.1% ± 3.3% | 24.9% ± 2.9% | 28.0% ± 3.0% | **3.39 [2.38–4.90]** | **3.97 [2.77–5.68]** | **4.21 [2.76–6.43]** | **4.66 [3.12–6.96]** |
| 2 outlets (n = 116) | 51.1% ± 4.6% | 18.9% ± 3.6% | 30.0% ± 4.2% | **2.38 [1.41–4.00]** | **3.93 [2.50–6.18]** | **2.88 [1.58–5.27]** | **3.49 [2.08–5.88]** |
| 3 or more outlets (n = 205) | 53.7% ± 3.5% | 18.0% ± 2.7% | 28.3% ± 3.1% | **2.15 [1.42–3.25]** | **3.52 [2.44–5.06]** | **1.94 [1.23–3.07]** | **2.91 [1.92–4.39]** |
| **1,600 m** | | | | | | | |
| None (n = 1,175) | 76.7% ± 1.2% | 11.3% ± 0.9% | 12.0% ± 0.9% | 1 (Ref.) | 1 (Ref.) | 1 (Ref.) | 1 (Ref.) |
| 1 outlet (n = 216) | 44.0% ± 3.3% | 28.2% ± 3.0% | 27.8% ± 3.0% | **4.35 [3.01–6.29]** | **4.02 [2.79–5.80]** | **6.07 [3.98–9.26]** | **5.39 [3.56–8.16]** |
| 2 outlets (n = 148) | 56.9% ± 4.0% | 16.8% ± 3.1% | 26.2% ± 3.6% | **2.01 [1.24–3.25]** | **2.94 [1.93–4.46]** | **2.42 [1.39–4.22]** | **2.81 [1.74–4.54]** |
| 3 or more outlets (n = 217) | 54.9% ± 3.4% | 20.2% ± 2.7% | 24.9% ± 2.9% | **2.50 [1.69–3.69]** | **2.89 [2.00–4.16]** | **2.57 [1.65–3.99]** | **2.49 [1.65–3.76]** |

**Note:**
* Adjusted for the participant's sex, age, tobacco smoking status, marital status, income, religion, occupation, and educational level. Bold texts denote statistical significance at 95% level of confidence.

*facto* legalization, suggesting that there was an influence of social desirability on survey findings in the past and that the true prevalence of cannabis use was considerably higher than the reported.

Despite the main intention of the cannabis decree being the use of cannabis for medical purposes (*HFocus Reporters, 2023*), less than 10 percent of our participants indicated

**Table 3 Density of cannabis outlets within distances from participant's residence and cannabis use disorder (among current users only).**

| Distance and density | Non-hazardous use (0–7 points) | Hazardous use or possible CUD (Eight points or higher) | Crude OR (95% CI) | Adj. OR (95% CI)* |
|---|---|---|---|---|
| **400 m** | | | | |
| None (n = 136) | 97.8% ± 1.3% | 2.2% ± 1.3% | 1 (Ref.) | 1 (Ref.) |
| 1 outlet (n = 37) | 97.4% ± 2.6% | 2.6% ± 2.6% | 1.20 [0.12–11.85] | N/A** |
| 2 outlets (n = 38) | 97.5% ± 2.5% | 2.5% ± 2.5% | 1.16 [0.12–11.47] | N/A** |
| 3 or more outlets (n = 35) | 100.0% ± 0.0% | 0.0% ± 0.0% | N/A** | N/A** |
| **800 m** | | | | |
| None (n = 121) | 97.5% ± 1.4% | 2.5% ± 1.4% | 1 (Ref.) | 1 (Ref.) |
| 1 outlet (n = 48) | 97.9% ± 2.1% | 2.1% ± 2.1% | 0.83 [0.08–8.20] | 1.42 [0.08–24.34] |
| 2 outlets (n = 32) | 93.9% ± 4.1% | 6.1% ± 4.1% | 2.53 [0.41–15.76] | 5.73 [0.28–116.91] |
| 3 or more outlets (n = 48) | 100.0% ± 0.0% | 0.0% ± 0.0% | N/A | N/A |
| **1,200** | | | | |
| None (n = 120) | 97.5% ± 1.4% | 2.5% ± 1.4% | 1 (Ref.) | 1 (Ref.) |
| 1 outlet (n = 57) | 98.3% ± 1.7% | 1.7% ± 1.7% | 0.69 [0.07–6.80] | N/A** |
| 2 outlets (n = 27) | 96.4% ± 3.5% | 3.6% ± 3.5% | 1.46 [0.15–14.59] | N/A** |
| 3 or more outlets (n = 47) | 97.9% ± 2.0% | 2.1% ± 2.0% | 0.82 [0.08–8.04] | N/A** |
| **1,600 m** | | | | |
| None (n = 119) | 97.5% ± 1.4% | 2.5% ± 1.4% | 1 (Ref.) | 1 (Ref.) |
| 1 outlet (n = 54) | 98.2% ± 1.8% | 1.8% ± 1.8% | 0.73 [0.07–7.13] | N/A** |
| 2 outlets (n = 32) | 97.0% ± 3.0% | 3.0% ± 3.0% | 1.21 [0.12–12.08] | N/A** |
| 3 or more outlets (n = 45) | 97.9% ± 2.1% | 2.1% ± 2.1% | 0.85 [0.09–8.34] | N/A** |

Notes:
* Adjusted for the participant's sex, age, tobacco smoking status, marital status, income, religion, occupation, educational level, and age of onset of cannabis use.
** Output could not be calculated due to unavailability of comparison groups.

medical purpose as the main reason for use. Such a phenomenon is similar to the assessment of the impact of medical cannabis legalization on recreational cannabis use in the United States (*Chiu et al., 2021*; *Wilkinson et al., 2016*). Interestingly, nearly three-fifths of our participants reported ingestion as the primary method of use (either by edible or infusion), which differed from the common methods of use in Canada (*Government of Canada, 2022*) and the United States (*Streck et al., 2019*).

The reason behind this high prevalence could be cultural: cannabis leaves were used, albeit discreetly, as a flavor enhancer in food (*Narongvit et al., 2021*), although historical sources and temple artworks also commonly included the use of smoked cannabis (*Sarakadee Lite Editors, 2023*; *Silpa Magazine Editors, 2023*). Unfortunately, we collapsed eating and drinking into the same category, which limited the ability to contextualize consumption. Future studies should consider separating the use of cannabis as food *vs.* drinks in order to understand consumption in a clearer manner.

We found associations between cannabis outlet density and lifetime and current cannabis use, but not hazardous/disorder cannabis use or amount of cannabis smoked (among cannabis smokers). The findings regarding prevalence of cannabis use were similar to that of a previous study among adolescents in California (*O'Hara, Paschall & Grube, 2023*) and youths in the state of Washington (*Rhew et al., 2022*). On the other hand, the

**Table 4 Cannabis use amount (grams per person per day of use) within past year among participants by density of cannabis outlets (among current users who smoked the cannabis) (n = 84).**

| Distance and Density | Dosage (mean ± SD) | Crude Beta (95% CI) | Adj. Beta (95% CI)* |
|---|---|---|---|
| **400 m** | | | |
| None (n = 28) | 1.22 ± 0.28 | 1 (Ref.) | 1 (Ref.) |
| 1 outlet (n = 12) | 1.77 ± 0.39 | 0.56 [−0.37 to 1.48] | 0.19 [−0.87 to 1.25] |
| 2 outlets (n = 15) | 2.69 ± 0.91 | 1.47 [−0.38 to 3.33] | 1.98 [−0.52 to 4.48] |
| 3 or more outlets (n = 16) | 1.02 ± 0.28 | −0.20 [−0.96 to 0.57] | 0.25 [−1.07 to 1.58] |
| **800 m** | | | |
| None (n = 27) | 1.13 ± 0.23 | 1 (Ref.) | 1 (Ref.) |
| 1 outlet (n = 11) | 1.77 ± 0.43 | 0.64 [−0.31 to 1.59] | −0.22 [−1.43 to 1.00] |
| 2 outlets (n = 14) | 2.64 ± 0.98 | 1.51 [−0.46 to 3.48] | 2.13 [−0.23 to 4.49] |
| 3 or more outlets (n = 20) | 1.19 ± 0.25 | 0.06 [−0.59 to 0.72] | 0.44 [−0.77 to 1.66] |
| **1,200 m** | | | |
| None (n = 28) | 1.36 ± 0.29 | 1 (Ref.) | 1 (Ref.) |
| 1 outlet (n = 13) | 1.66 ± 0.38 | 0.30 [−0.64 to 1.23] | −1.00 [−2.57 to 0.57] |
| 2 outlets (n = 14) | 2.53 ± 0.99 | 1.17 [−0.85 to 3.19] | 1.36 [−0.68 to 3.40] |
| 3 or more outlets (n = 18) | 1.14 ± 0.15 | −0.22 [−0.87 to 0.43] | −0.15 [−1.21 to 0.92] |
| **1,600 m** | | | |
| None (n = 33) | 1.11 ± 0.21 | 1 (Ref.) | 1 (Ref.) |
| 1 outlet (n = 11) | 1.93 ± 0.39 | 0.82 [−0.06 to 1.70] | 0.27 [−0.91 to 1.44] |
| 2 outlets (n = 12) | 3.00 ± 1.11 | 1.89 [−0.33 to 4.10] | 2.05 [−0.12 to 4.22] |
| 3 or more outlets (n = 19) | 1.07 ± 0.16 | −0.04 [−0.56 to 0.48] | 0.17 [−0.80 to 1.15] |

Note:
* Adjusted for the participant's sex, age, tobacco smoking status, marital status, income, religion, occupation, educational level, and age of onset of cannabis use.

findings regarding hazardous use differed from a study among young adults in California (*Pedersen et al., 2021*), which found a positive association. In our study findings, there seemed to be a dose-response relationship between the number of outlets within a 400-m radius and cannabis use behaviors. However, the relationship was mirror-imaged J-shape (*i.e.*, stronger association for 1 outlet in the given distance and weaker associations for a higher number of outlets) at other distances.

One potential source of bias in the study findings was our re-categorization of those who answered "Don't know" with regard to cannabis outlet density as those who reported no cannabis outlets within a given distance from their home. Those who answered "Don't know" were more likely to be never users and less likely to be former and current users, thus recoding "Don't know" as "No outlets" placed more never users and fewer former and current users in the "No outlets" group and biased the association between outlet density and cannabis use behaviors away from the null. We are not sure how the recoding of outlet density biased the association between outlet density and amount of cannabis used (in grams) toward the null value. However, the relative consistency between the two models (with and without imputation of age at initiation of cannabis use) suggested that residual confounding by imputation of this covariable was relatively small. One additional source of

bias is that although we collected data from all adults in a given household, we lacked the oversight to issue household-level unique identification number, and thus we could not account for clustering at the household level in our analyses. Cannabis use behaviors may cluster within a household in a similar manner to other health behaviors (*Niermann, Spengler & Gubbels, 2018*). Non-adjustment of clustering at the household level could have introduced bias to our findings of cannabis use behaviors being unevenly distributed between households with more and fewer participants.

We do not deem our study findings to have adequate internal consistency to support the study hypotheses regarding the positive association between outlet density and cannabis use behaviors and hazardous cannabis use, cannabis use disorder, or dosage of cannabis (among cannabis smokers). However, a number of issues need to be taken into account regarding our measurement of cannabis outlet density. In our analyses, we found a small number of contradictions between responses regarding outlet density (for example, a participant might have reported having no outlet within an 800 m radius, but reported 1 or 2 outlets within the 400 m radius). We decided to present the data as they appeared, as imposing the value of 0 onto the number of outlets at shorter radii may in itself have biased the estimation downward, and there was no method to triangulate each response and determine the actual presence or absence of outlets within a given radius. Had we used a fixed algorithm for data cleaning with *a priori* assumptions, the findings of this study could have been different from the existing observation. In addition, distance itself does not imply ease of access. We did not collect data regarding the amount of cannabis available at each outlet, the opening hours, or other factors affecting access. Future studies should consider making a thorough and comprehensive calculation of outlet density and measuring ease of access at each outlet in addition to the outlet density itself. Furthermore, our study findings could have been affected by unmeasured psychological trauma and mental health disorders. A systematic review found an association between cannabis dependence and stress factors, critical life events, and comorbid disorders (*Schlossarek et al., 2016*). Considering that stress and critical life events are attributable to mental disorders, and the strong linkage between mental disorders and substance use disorders (*Muñoz-Galán et al., 2023*; *Urits et al., 2020*), the potential confounding by comorbidities should be considered in the interpretation of the study findings. Furthermore, future studies should also include comorbidities as part of the measurement of participant characteristics.

## Strengths and limitations

The primary strengths of our study were the representative nature of our study data, and the post-decriminalization context which might have eased the influence of social desirability on our study findings. However, a number of limitations should also be considered in the interpretation of our study findings. Firstly, we did not apply an algorithm or triangulate information regarding outlet density at different distances, which could have yielded different results. Secondly, the cross-sectional study design did not allow for the ascertainment of temporality. Considering the recent growth in the number of cannabis outlets in Thailand, the study outcomes could have preceded the exposure in

this study. Lastly, our study participants were adults aged 20 years or older. Our study findings could not be generalized to other populations of interest, *e.g.*, adolescents aged under 20 years.

## CONCLUSIONS

In this nationally-representative survey, described cannabis use behaviors and health outcomes, and the extent that these behaviors and outcomes were associated with density of cannabis outlets. we found that 15 percent of adults in Thailand were current cannabis users, the majority of whom used cannabis by ingestion and for recreational purpose. However, cannabis use disorder was uncommon among current users. Cannabis outlet density was associated only with the probability of being former and current cannabis users. Limitations with regard to outlet density measurement and lack of temporality should be considered as caveats in the interpretation of the study findings.

## ACKNOWLEDGEMENTS

We wish to thank all study participants for their valuable time and all members of the data collection team for their tireless efforts.

### Funding
This work was supported by Thailand's Health Systems Research Institute (No. HSRI 66-027). The funders had no role in study design, data collection and analysis, decision to publish, or preparation of the manuscript.

### Grant Disclosures
The following grant information was disclosed by the authors:
Thailand's Health Systems Research Institute: HSRI 66-027.

### Competing Interests
The authors declare that they have no competing interests.

### Author Contributions

- Wit Wichaidit conceived and designed the experiments, performed the experiments, analyzed the data, prepared figures and/or tables, and approved the final draft.
- Ilham Chapakiya analyzed the data, prepared figures and/or tables, and approved the final draft.
- Aneesah Waeuseng analyzed the data, prepared figures and/or tables, and approved the final draft.
- Kemmapon Chumchuen analyzed the data, authored or reviewed drafts of the article, and approved the final draft.
- Sawitri Assanangkornchai conceived and designed the experiments, authored or reviewed drafts of the article, and approved the final draft.
## Human Ethics

The following information was supplied relating to ethical approvals (*i.e.*, approving body and any reference numbers):

This study received ethical approval from the Human Research Ethics Unit, Faculty of Medicine, Prince of Songkla University (REC. 65-434-19-2).

## Data Availability

The data necessary to replicate the study findings, the translation of the study instrument sections necessary for replication of the study findings and the R codes are in the Supplemental Files.

## Supplemental Information

Supplemental information for this article can be found online at http://dx.doi.org/10.7717/peerj.17317#supplemental-information.

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
