# Peer review of "Density of cannabis outlets *vs.* cannabis use behaviors and prevalent cannabis use disorder: findings from a nationally-representative survey"

_PeerJ, doi:10.7717/peerj.17317_

## Round 0.1 · original submission · Major Revisions

Thank you for your patience. Please note the comments from the reviewers and pay particular attention to the Results section.

·

Basic reporting

no comment

Experimental design

This study is novel and original. The knowledge gap is well-defined. Rigorous investigation was performed. However, the methods were not described in sufficient detail. Missing data were not properly handled, and sensitivity analyses are required.

- In lines 194 and 197. Please provide more details on the descriptive statistics. For example, “we described cannabis use behaviors and prevalence of hazardous cannabis use and cannabis. Means and standard deviations were used to describe continuous variables and frequencies and proportions were used to describe categorical variables.” In line 196, please elaborate on what parameters were calculated from the logistic regression. For example, “odds ratios and their 95% confidence intervals were computed from the logistic regression models.” Also please indicate what criteria were used to determine a statistical significance. For example, “a two-sided p-value was used to indicate statistical significance.”

- It is unclear how many members were recruited per household. Only one member or all the members who have met the inclusion criteria? Could you provide the minimum, maximum, mean, or median number of participants clustered within the same household? Could the household be confounding effects of the association between cannabis outlet density and former and current use of cannabis, or the association between outlet density and hazardous or harmful use of cannabis or cannabis use? Have you tried adjusting for households in the logistic regression models?

- In lines 211-214, you mentioned that half of the participants answered “Don’t know” to the number of outlets, and their responses were recoded to 0. This seems to be a very high missing rate, especially for such an import variable. Simply recoding the “don’t know” responses to 0 could introduce misclassification and bias the study results. I would assume that most of the people who reported “don’t know” are never users, which is why they don’t know the number of outlets. In this case, recoding their responses to 0 would bias the association between cannabis outlet density and former and current use of cannabis away from the null. So the significant association you observed might be just caused by the recoding. Could you conduct a sensitivity analysis to ensure that that’s not the case? For example, you could compare the prevalence of former/current use/never smokers among those who reported the number of outlets vs. those who reported “don’t know” to see if there are significant differences. You could also impute missing data. If there are study participants who reported the number of outlets living in the same household with those who reported “don’t know”, you could recode “don’t know” to those valid responses provided by people living in the same household. If no valid response is provided for that household, you could use the responses from the nearest households.

- In lines 216-220, you mentioned that participants who didn’t report their age of onset of cannabis were imputed with the median age of initiation (25 years). Could you fit the logistic regression in Tables 3 and 4 among those who reported their age of onset. If the sample size is an issue, you could only provide the crude beta with the age of onset being the extra covariate. Could you also check if the three variables reported outlet density, cannabis use amount, and cannabis use disorder are significantly different in those who reported age of onset vs. those who did not report? If they are, there might be some patterns in those missing data. For example, some people have an early onset might feel ashamed to report their age of onset, so their true onset age might be earlier than 25 years. Thus, the median age of initiation might not be sufficiently controlled for in regression models by imputing the missing age to 25 years. There might be residual confounding. This could partly contribute to the null association and should be discussed.

Validity of the findings

The authors stated that they found associations between cannabis outlet density and lifetime and current cannabis use, but not hazardous/disorder cannabis use or amount of cannabis smoked (among cannabis smokers). Given that the missing data were not properly handled, I don’t think the current study results provide sufficient support for their conclusions.

Additional comments

no comment

Reviewer 2 ·

Basic reporting

The article is clear, succinct, and unambiguous. The authors have excellent writing skills.

The literature review is very good with up-to-date references.

The article structure is very good and tables of raw data are included.

Yes, the results are relevant to the hypotheses.

Experimental design

Yes, this is original primary research that meets the aims and scope of the journal.

The research questions are well-defined. It is important to add to the literature on cannabis use in different countries.

I am impressed by the detail the authors used to describe their rigorous investigation. The study design is comprehensive and the authors engaged a survey research firm. They gave examples of the questions. The authors showed sensitivity toward respondents who were not able to read. The sample size of 2191 is strong.

Yes, the methods were sufficiently described to allow for replication.

Validity of the findings

I am not a research methodologist/statistician and hope another reviewer has this expertise and can be helpful here.

Cannabis outlet density is an issue in every country and this article contributes to the expansion of knowledge and gives good suggestions for future research.

The Strengths and Limitations section of the article is very strong.

Additional comments

Overall, this is a very good article. It gives a good picture of cannabis use in Thailand. It supports the need for more research on the relationship between access to cannabis and its use.

I have no concerns regarding the article.

·

Basic reporting

Abstract and Background:
The abstract provides a clear overview of the study’s objectives and
context. It highlights the recent decriminalization of cannabis in
Thailand and the lack of nationally representative data on cannabis use
behaviors and their association with the cannabis outlet density.

The background sets the stage for the study by stating the objectives
clearly. However, it could benefit from a more detailed explanation of
why these particular objectives were chosen and how they relate to the
gaps in this study.

The objectives of this study were to describe the prevalence of cannabis use behaviors and cannabis use disorders among the general adult population in Thailand and to describe the extent to which the density of cannabis outlets is associated with cannabis use behaviors, cannabis use disorder, and the amount of cannabis smoked per day. These objectives are clear, of scope, and justified in the introductory section. The hypotheses of this study are not evident.

English is clear and professional, but spaces between words are recommended. (e.g line57, 285). Check when a word should be plural (e.g., line 67) disorder/disorders) Please note this recommendation for the entire document.
You should thoroughly review the spaces and some wording. (e.g line 61).

The references have been updated and are in accordance with the subject matter; however, there is a high percentage of references to books and reports. In Table 1, the footnote has been revised accordingly. Generally, the tables are correct, relevant, and illustrative.

Experimental design

Methods:

The study design is appropriate for the research question, and the inclusion of both literate participants who self-administered the questionnaire and those who were interviewed is commendable for its inclusivity.

However, the text does not specify the sampling method used to select the 11 provinces and the Bangkok Metropolitan Area, which raises questions regarding the representativeness of the sample.

The method section could be better organized, and it could be distributed into subsections that are commonly found in studies. This part needs to be significantly improved (Design, Participants, Instruments, Participants, Data analysis) and should be clearly reflected. It is not advisable to incorporate results in the method, or at least it is unusual and could generate confusion.

The use of descriptive statistics and multivariate logistic regression analyses is standard; however, the text lacks details on the specific statistical methods and any adjustments made for potential confounders.

Validity of the findings

Methods:

The study design is appropriate for the research question, and the inclusion of both literate participants who self-administered the questionnaire and those who were interviewed is commendable for its inclusivity.

However, the text does not specify the sampling method used to select the 11 provinces and the Bangkok Metropolitan Area, which raises questions regarding the representativeness of the sample.

The method section could be better organized, and it could be distributed into subsections that are commonly found in studies. This part needs to be significantly improved (Design, Participants, Instruments, Participants, Data analysis) and should be clearly reflected. It is not advisable to incorporate results in the method, or at least it is unusual and could generate confusion.

The use of descriptive statistics and multivariate logistic regression analyses is standard; however, the text lacks details on the specific statistical methods and any adjustments made for potential confounders.

Results:

The results indicated a significant association between the number of cannabis outlets and the odds of current cannabis use, which is an important finding.
However, the text was cut off before providing full details on the results, making it impossible to evaluate the completeness of the reported findings. This section provides detailed evidence to answer the two research questions and objectives.
The stacking between numbers should be improved in the results table.
The conclusion is somewhat reiterative, but I could take the opportunity to summarize more about the two stated objectives

Additional comments

Evaluation:

Suitable for publication: This study addresses a timely and relevant question, and the methods used are generally appropriate.
Needs substantial modifications: To reach the Q1 journal level, the article requires more detail on the sampling method, statistical analyses, and full results. The discussion section, which is not included in the selected text, should also critically evaluate the findings in the context of the existing literature and address the study’s limitations.

Not suitable: As it stands, the incomplete results section makes the article unsuitable for publication. In summary, the article has the potential to contribute valuable insights into
cannabis use behaviors in Thailand post-decriminalization but requires
significant revisions to meet the rigorous Q1 standards.

---

## Round 0.2 · accepted · Accept

Thank you for addressing the reviewers' comments.

·

Basic reporting

no comments

Experimental design

my previous comments have been sufficiently addressed

Validity of the findings

my previous comments have been sufficiently addressed